# Panobinostat Synergistically Enhances the Cytotoxicity of Microtubule Destabilizing Drugs in Ovarian Cancer Cells

**DOI:** 10.3390/ijms232113019

**Published:** 2022-10-27

**Authors:** María Ovejero-Sánchez, Gloria Asensio-Juárez, Myriam González, Pilar Puebla, Miguel Vicente-Manzanares, Rafael Pélaez, Rogelio González-Sarmiento, Ana Belén Herrero

**Affiliations:** 1Institute of Biomedical Research of Salamanca (IBSAL), 37007 Salamanca, Spain; 2Molecular Medicine Unit, Department of Medicine, University of Salamanca, 37007 Salamanca, Spain; 3Institute of Molecular and Cellular Biology of Cancer (IBMCC), University of Salamanca-CSIC, 37007 Salamanca, Spain; 4Laboratorio de Química Orgánica y Farmacéutica, Departamento de Ciencias Farmacéuticas, Facultad de Farmacia, Universidad de Salamanca, 37007 Salamanca, Spain; 5Centro de Investigación de Enfermedades Tropicales de la Universidad de Salamanca (CIETUS), Facultad de Farmacia, Universidad de Salamanca, 37007 Salamanca, Spain

**Keywords:** panobinostat, microtubule-destabilizing agents, tubulin, acetylation

## Abstract

Ovarian cancer (OC) is one of the most common gynecologic neoplasia and has the highest mortality rate, which is mainly due to late-stage diagnosis and chemotherapy resistance. There is an urgent need to explore new and better therapeutic strategies. We have previously described a family of Microtubule Destabilizing Sulfonamides (MDS) that does not trigger multidrug-mediated resistance in OC cell lines. MDS bind to the colchicine site of tubulin, disrupting the microtubule network and causing antiproliferative and cytotoxic effects. In this work, a novel microtubule-destabilizing agent (**PILA9**) was synthetized and characterized. This compound also inhibited OC cell proliferation and induced G2/M cell cycle arrest and apoptosis. Interestingly, **PILA9** was significantly more cytotoxic than MDS. Here, we also analyzed the effect of these microtubule-destabilizing agents (MDA) in combination with Panobinostat, a pan-histone deacetylase inhibitor. We found that Panobinostat synergistically enhanced MDA-cytotoxicity. Mechanistically, we observed that Panobinostat and MDA induced α-tubulin acetylation and that the combination of both agents enhanced this effect, which could be related to the observed synergy. Altogether, our results suggest that MDA/Panobinostat combinations could represent new therapeutic strategies against OC.

## 1. Introduction

Ovarian cancer (OC) is the most lethal gynecologic neoplasia, causing around 210,000 annual deaths worldwide [1]. High mortality is mainly due to late diagnosis because tumors grow with non-specific clinical manifestations [2,3]. Moreover, relapses and the development of chemoresistance are common. Around 75% of patients diagnosed in advanced stages will relapse within the first 2 years after initial treatment [3,4,5,6]. Therefore, there is a clear need to develop new therapeutic strategies, such as efficient drug combinations, that might prevent the development of drug resistance and tumor relapse, and improve OC survival.

Microtubules (MTs) are polymers composed of 13 protofilaments that extend along the same axis forming bundled, cylinder-shaped structures. The building block of microtubules is tubulin, which forms head-to-tail interactions in a GTP-dependent manner. There are two major isoforms of tubulin, α- and β-. These isoforms form heterodimers that grow into fully formed microtubules. Microtubules play an essential role in several cellular processes such as division and cellular motility [7,8]. Their ability to form the mitotic spindle during division makes microtubules a target for the development of chemotherapeutic drugs. Microtubule-targeting agents are among the first forms of chemotherapy for the treatment of several tumors, including OC. These compounds can be divided into two main groups: microtubule-destabilizing agents and microtubule-stabilizing agents. Microtubule-destabilizing agents (MDA) inhibit microtubule polymerization and include several compounds such as the Vinca alkaloids, colchicine, or combretastatins [9]. On the other hand, microtubule-stabilizing agents inhibit microtubule depolymerization, and its main representative family is taxanes [8].

OC standard treatment includes a primary cytoreductive surgery of the tumor and platinum/taxane-based chemotherapy [10,11]. Taxane-based chemotherapy uses taxanes, such as paclitaxel or docetaxel, that bind to the hydrophobic taxoid site of β-tubulin [12]. These compounds arrest cells in G2/M, decrease microtubule depolymerization, and ultimately inhibit cancer cell division [12,13]. However, paclitaxel efficiency decreases or disappears in tumor cells with the development of chemoresistance. This resistance results from changes in the expression or posttranslational modifications of tubulin proteins, altered expression of certain drug transporters, such as P-gp, modifications of the levels of cell cycle-related proteins, or variation of several cellular processes (autophagy, oxidative stress, or microRNAs deregulation) [3,12]. Thus, the development of novel microtubule-binding agents with antitumor activity remains a priority. In this regard, we have previously reported a new family of Microtubule Destabilizing Sulfonamides (MDS) that mostly avoid multidrug-mediated resistance and exhibit improved aqueous solubility. These compounds bind to the colchicine site of tubulin, disrupting the microtubule network and presenting antiproliferative and cytotoxic effects in OC cell lines [14].

Histone deacetylases (HDACs) exert an essential role in epigenetic regulation, mainly acting as transcriptional repressors. Overexpression of HDACs is commonly observed in several forms of cancer, including OC [15,16,17]. HDACs overexpression in OC is related to tumor progression, poor prognosis, and the development of chemoresistance [15,16,17]. Therefore, histone deacetylase inhibitors (HDACi) represent promising agents for OC treatment. These compounds inhibit HDACs by promoting the transcriptional activation of specific genes repressed by the tumor program [18]. Pan-histone deacetylase inhibitors, such as Panobinostat (LBH) or Vorinostat, inhibit several HDACs, including histone deacetylase 6 (HDAC6) [19]. In addition to its effect on histones, HDAC6 also deacetylates Lys(K)40 of α-tubulin [7]. Therefore, HDACi restore (or increase) α-tubulin acetylation. Some reports have shown that HDACi synergistically enhanced the cytotoxic effect of paclitaxel. This higher cytotoxicity was related to an increase in apoptosis or in tubulin acetylation [20,21,22,23,24]. Besides this, several studies have shown that novel HDAC/tubulin dual inhibitors present a strong antitumor and antiangiogenic potency [25,26,27,28,29,30,31].

In this study, we explored whether LBH could enhance the cytotoxic effect of microtubule-destabilizing agents. For this purpose, we combined several MDS previously designed and synthesized in our laboratory, with LBH and evaluated their cytotoxicity in OC cells. We observed that LBH and the three MDS used exerted a synergistic cytotoxic effect in all four different OC cell lines studied. The three MDS used here share a common diarylsulfonamide structure that could cause off-target effects unrelated to their effect on MTs. To resolve this decisively and show that the effects are largely independent of the type of chemical scaffold present in the MDS, we synthesized a new microtubule destabilizing agent, **PILA9**, with an indolecombretastatin structure, structurally very different from the diarylsulfonamides [32]. We found that **PILA9** inhibited cell proliferation, induced G2/M cell cycle arrest, and induced apoptosis at doses much lower than the rest of the MDS used. Moreover, this new compound also synergized with Panobinostat in OCCLs. Mechanistically, we found that LBH and the MDA induced α-tubulin acetylation and that the combination of LBH with these compounds enhanced this effect. Together, these data suggest that the combined effect of MDA/LBH could have an important preclinical basis for future clinical testing.

## 2. Results

### 2.1. Panobinostat Enhances Cytotoxicity of Microtubule-Destabilizing Sulfonamides in OCCLs

Several reports have revealed that HDAC inhibitors enhanced the cytotoxicity of tubulin-interacting drugs [20,21,22,23,24,25,26,27,28,29,30,31]. Based on these studies, we analyzed the combined cytotoxic effect of LBH and three MDS previously designed and synthesized in our laboratory (**38**, **42**, or **45**) [14] on several OC cell lines. For this purpose, we performed apoptosis assays using different concentrations of the compounds (Figure 1, Figure 2 and Figure 3). Cell survival after combined treatments was lower than that observed with each drug individually. To determine the type of interaction between the drugs, combination indices (CI) were calculated using the Compusyn Software. CIs were less than 1 in the different combinations analyzed, which reveals that LBH and the MDS used are synergic in terms of cytotoxicity on these cell lines.

### 2.2. The Compound **PILA9** Inhibits Cell Proliferation, Induces G2/M Cell Cycle Arrest and Apoptosis, and Synergizes with Panobinostat in OCCLs

Next, we proposed to study the effect of LBH with another MDA (**PILA9**), which has a different chemical structure to the other MDS used throughout. MDS share a common diarylsulfonamide structure and sulfonamides are privileged scaffolds able to bind very diverse targets. To ensure that the observed effects are related to the effect of these drugs on tubulin, we selected a *Z*-stilbene, an analog of combretastatin A4 with a *Z* olefin bridge instead of the sulfonamide and a 3-substituted indole ring replacing the mono- or di-substituted phenyl ring of MDS. We conserved the trimethoxyphenyl ring as a structural requirement for strong binding to the colchicine-binding site on tubulin and potent cytotoxic activity (Appendix A). Ensemble molecular docking studies for compound **PILA9** at the colchicine site of tubulin suggest a similar binding mode to that of combretastatin A4 (CA4) (Figure 4A). **PILA9** binds to zones A and B of the colchicine site in a similar disposition to combretastatin A4, with a very similar arrangement of the two phenyl rings: the trimethoxyphenyl ring of both compounds sits in the A zone, and the other aromatic system in zone 2, equally to the MDS. A close overlap of the trimethoxyphenyl ring of **PILA9** with that of the X-ray structure of combretastatin A4 in complex with tubulin is observed. The trimethoxyphenyl ring inserts edgewise toward the surface of sheets S8 and S9 between the sidechains of Ala316β, Val318β, and Ala354β, and covered by helices H7 and H8 and by the H7-H8 loop, contacting the sidechains of Cys241β, Leu242β, Leu248β, Ala250β, and Leu255β. The olefinic bridge is also placed similarly to that of combretastatin A4 and the sulfonamide bridges of the MDS, packed against helix H8 between Leu255β and Leu248β in a hydrophobic pocket at the interdimer interface. The indole system overlaps as well with the phenyl ring of combretastatin A4 and the MDS, with the N-methyl replacing the methoxy groups of combretastatin A4 or the MDS. The indole ring lays behind helix H8, making carbonyl pi interactions with Asn258β and above the sidechains of Ala316β and the methylene groups of the sidechain of Lys352β. The carbonyl group of the carbamoyl group hydrogen bonds to the backbone NH of Val181α, in a similar way as the hydroxyl group of combretastatin A4 or the ketone of the tropolone of colchicine, while the amino group hydrogen bonds to the backbone carbonyl of Asn349β. The similar binding modes to the colchicine site of tubulin of **PILA9**, the MDS, and combretastatin A4 or colchicine [9] suggest a common mechanism of action mediated by tubulin binding.

Before carrying out the combination study, we decided to test the antitumor activity of **PILA9** and its effect on the microtubule network. For this purpose, OCCLs were treated with different concentrations of **PILA9** for 24, 48, and 72 h; then cell viability was measured by MTT. A dose- and time-dependent anti-proliferative effect was observed in the four cell lines analyzed, with A2780 and OVCAR-8 being the two most sensitive cell lines (Figure 4B). It is noteworthy that this compound exerted a strong anti-proliferative activity, with IC_50_ values that ranged from 1.37 nM for A2780 up to 6.43 nM for IGROV-1, much lower than those reported for the MDS **38**, **42,** and **45** (from 7 nM to 492 nM, depending on the compound and the cell line used) [14] (Table 1). To further characterize the anti-proliferative activity of **PILA9**, we studied its effect on the cell cycle. We found that **PILA9** caused an accumulation of cells in the G2 phase in all cell lines tested, and a marked increase in the percentage of dead cells (*p*-value < 0.05) (Figure 4C), except for A2780 at the doses and conditions employed. This increase in the sub-G0 phase indicated that **PILA9** produced a strong cytotoxic effect.

Next, we analyzed whether the observed cell death was due to the induction of apoptosis. We found that the percentage of apoptotic cells (annexin V/PI positive cells) increased in the presence of **PILA9**, as shown in Figure 4D. Since **PILA9** is expected to bind tubulin, we next test whether **PILA9** had structural effects on the microtubule network. For this purpose, A2780 and SK-OV-3 cells were treated with **PILA9,** and 24 h later, α-tubulin was observed by immunofluorescence. We found that cells treated with **PILA9** exhibited a more diffuse distribution of α-tubulin than untreated cells (Figure 4E), consistent with a depolymerizing effect similar to that of other colchicine-site binding compounds [33].

Next, we compared the effect of the combination of LBH and **PILA9** with that observed by each drug in monotherapy. As shown in Figure 5, the percentage of live cells was much lower in the combined treatments in the four OCCLs analyzed. To determine the type of interaction between these two compounds, combination indices were calculated using Compusyn software. All the CIs were well below 1 in the four OCCLs analyzed, revealing a synergistic interaction between the two drugs (Figure 5). Interestingly, IGROV-1 and SK-OV-3 displayed very low CIs.

### 2.3. Microtubule-Destabilizing Agents and Panobinostat Induce the Acetylation of α-Tubulin in OCCLs

HDACi produces diverse cellular effects, one of the most prominent being the induction of α-tubulin acetylation [18]. It has been described that HDACi activity on histone acetylation occurred during the first 24 h of HDACi addition [18]. Consequently, we decided to study the effect of LBH on this α-tubulin posttranslational modification in OC cells after 24 h of LBH treatment. Levels of total and acetylated α-tubulin were measured by western blot. As shown in Figure 6A, LBH treatment increased tubulin acetylation in the four OCCLs analyzed, as expected.

Next, we decided to analyze the effect of the MDA on tubulin acetylation. Compound **38** and **PILA9** were selected as representative MDA and tested. OC cells were treated with different doses of these compounds and the levels of tubulin acetylation were calculated after each treatment. We found that the MDA agents induced α-tubulin acetylation at low doses, whereas at higher doses the proportion of tubulin acetylated decreased, especially in the case of **PILA9** (Figure 6B,C), similar to the reported effect of colchicine-related compounds [34].

### 2.4. Cotreatment with MDS and Panobinostat Induces a Stronger Acetylation of α-Tubulin

Finally, we decided to analyze the effect of the combined treatment of MDS and LBH on tubulin acetylation. For this purpose, OC cells were treated with LBH and/or **38**/**PILA9** for 24 h. As shown in Figure 7, an increase in the acetylation of α-tubulin was observed by flow cytometry after LBH or **38** individual treatments and especially after co-treatments compared to untreated cells. These results were confirmed by immunofluorescence; as shown in Figure 7, LBH-treated cells showed higher levels of acetylated tubulin (green) that was located through the cytoplasm. In the case of **38**-treated cells, marked acetylation was detected around the nuclei. When cells were treated with both compounds, higher levels of acetylation were observed compared to each individual treatment. We next addressed whether these effects were also driven by the combination of LBH with MDS **42**, **45**, or **PILA9**. We detected an increase in α-tubulin acetylation in response to the different combinations of MDS and LBH by western blot and immunofluorescence (Figure 8 and Appendix A), except for OVCAR-8 cells treated with LBH-**42** at the conditions assayed.

## 3. Discussion

Conventional therapy for ovarian cancer includes tumor cytoreductive surgery, typically followed by platinum/taxane-based chemotherapy [10,11]. Taxanes bind to the hydrophobic taxoid site of β-tubulin inducing microtubule stabilization, G2/M cell cycle arrest, and inhibiting cell division [12]. However, its efficacy is limited, which is due to low aqueous solubility, elevated toxicity at high doses, and the appearance of chemoresistance [3,12]. To increase the potential of taxanes and other tubulin-interacting compounds, their combination with several antitumor drugs, such as HDACi, is being explored [20,21,22,23,24,25,26,27,28,29,30,31,35,36,37]. Here, we describe for the first time the combination of Microtubule-Destabilizing Sulfonamides (MDS) with Panobinostat (LBH), an HDACi. Such combinations produce a synergistic effect in OC cells in terms of cytotoxicity. In addition, we describe a novel compound structurally derived from combretastatin A4, **PILA9**. **PILA9** displayed an extraordinary cytotoxic effect against OC cells. This effect was also increased when **PILA9** was combined with LBH.

Several preclinical studies have shown that HDACi increase the cytotoxic effects of microtubule-binding agents such as paclitaxel [20,21,22,23,24,36]. Based on these reports, several dual HDAC/tubulin inhibitors are considered promising against different tumor types [25,26,27,28,29]. In OC, the combined HDACi/paclitaxel treatment enhances the cytotoxic effect of the individual compounds, which has been ascribed to a cooperative effect on microtubule stabilization through tubulin acetylation [20,21,23,24]. Indeed, those HDACi that target HDAC6 induce tubulin acetylation. HDAC6 deacetylates the K40 residue of α-tubulin [7,19]. In this study, we found that LBH treatment increased the ratio of acetylated tubulin to total tubulin in the four OC cell lines studied, which is consistent with the results from previous reports [23,24,31].

Microtubule-destabilizing agents, such as vincristine, combretastatin A4, ABT-751, or colchicine, have been employed for the treatment of different tumor types [31,38,39]. These compounds impair the microtubule network by disrupting their assembly, produce a G2/M cell cycle arrest, and induce apoptosis [14,38,39,40,41,42]. These effects are also produced by our previously described MDS [14] and by **PILA9**. Molecular docking studies predict the **PILA9** binds to the colchicine site of tubulin, which likely underlies the microtubule alterations observed after treatment with this compound. The binding mode of **PILA9** to the colchicine site of tubulin is like those of the MDS and to the experimentally determined binding modes of other colchicine site ligands such as the combretastatin A4 or colchicine [9], thus suggesting a common mechanism of action mediated by tubulin binding.

Some reports have also analyzed the effect of the combination of HDACi with microtubule-destabilizing drugs [31,37,43,44,45]. The authors showed that the addition of HDACi to eribulin or vincristine synergistically induced cytotoxicity on breast cancer, lymphoma, sarcoma, or leukemia cells [31,37,43,44,45]. The synergistic effect of HDACi and antimitotic agents has also led to the design of colchicine or isocombretastatin-HDACi hybrids [30,46]. These hybrids have both HDAC inhibitory activity and tubulin inhibitory activity, exhibiting powerful in vitro anti-proliferative effects on diverse cancer cell lines. Here we found that LBH enhanced the cytotoxicity of microtubule-destabilizing agents in the four OC cell lines analyzed, suggesting that the combination of different microtubule-destabilizing drugs and HDACi could also be explored for the treatment of OC. Moreover, we show that the interaction between these two types of drugs was synergic in all cases, with low combination indices, especially in the SK-OV-3 cell line.

The mechanism that explains the synergism of the microtubule-destabilizing drugs/HDACi combination has yet to be deciphered. In this regard, we found that the MDS analyzed and **PILA9** induced tubulin acetylation at low concentrations, and the combination of these microtubule-destabilizing drugs with LBH further enhanced the levels of tubulin acetylation. Conversely, it inhibits acetylation at high doses (Figure 6C). Consistently with our findings, some reports have described that colchicine analogs had different effects on microtubule dynamics and tubulin acetylation depending on the drug concentration or incubation times, increasing acetylation at low doses and decreasing it at high doses [34,47,48]. The explanation for this behavior may lie in the overall effect of the inhibitors at different doses. At low doses, the inhibitors have minor effects on the organization of the microtubules, but they may affect interaction with additional partners, for example, HDAC6. Conversely, at high doses, disassembly predominates, reducing acetylation as the amount of microtubules, that is, polymerized tubulin, decreases. It is important to highlight that acetylation only happens in the lumen of microtubules [49], contributing to the stabilization of the polymer by reducing the degree of freedom of the αK40 loop [50]. The mechanism of action of our microtubule-destabilizing agents might also differ at different doses or incubation times, which might correlate with the observed effects on tubulin acetylation. Interestingly, acetylated tubulin observed after treatment with the microtubule-destabilizing agents used in this work was located in the cytoplasm, but it was clearly accumulated around the cell nuclei. Such localization may be caused by a combination of increased actin retrograde flow due to the effect of microtubule depolymerization on the small GTPase RhoA, which increases contraction and accelerates retrograde flow [51,52], and a smaller size of microtubules [34], which would make them more amenable to actin-driven repositioning [53].

Some reports have described a correlation between the levels of tubulin acetylation and the degree of apoptosis [34,54,55,56]. Indeed, Wang et al. [54] described that when tubulin acetylation levels reach a “threshold value”, cells’ fate was cell death by apoptosis [54]. These findings could explain the higher apoptosis and tubulin acetylation found in the combined treatments. However, more experiments are needed to deep into the mechanism of action of the drugs used in this study and also to evaluate their efficacy in vivo.

## 4. Materials and Methods

### 4.1. Cell Lines and Culture Conditions

Ovarian cancer cell lines (OCCLs) IGROV-1, OVCAR-8, SK-OV-3, and A2780 were used in this work. OVCAR-8 and SK-OV-3 were obtained from the American Type Culture Collection (ATCC), A2780 from the European Collection of Authenticated Cell Cultures (ECACC), and IGROV-1 from Merck Millipore. A2780 and IGROV-1 were cultured in RPMI 1640 medium (Gibco, Waltham, MA, USA) supplemented with 10% heat-inactivated fetal bovine serum (FBS) (Gibco, Waltham, MA, USA) and 1% penicillin/streptomycin (Gibco, Waltham, MA, USA). OVCAR-8 and SK-OV-3 were cultured in Dulbecco’s modified Eagle’s medium (DMEM) (Gibco, Waltham, MA, USA) supplemented with 10% FBS and 1% penicillin/streptomycin. All cells were incubated at 37 °C in a 5% CO_2_ atmosphere. The presence of mycoplasma was routinely checked with the MycoAlert kit (Lonza, Basel, Switzerland) and only mycoplasma-free cells were used in the experiments.

### 4.2. Reagents

Panobinostat (LBH) was provided by Novartis Pharmaceuticals (Basel, Switzerland). Microtubule destabilizing sulfonamides (MDS) **38**, **42**, and **45** were previously synthesized by our group [14]. Indolecombretastatin **PILA9** (**P9**) was synthesized from (*Z*)-1-Methyl-5-(3,4,5-trimethoxystyryl)-1*H*-indole, synthesized as previously described [57].

### 4.3. Synthesis of (Z)-1-Methyl-5-(3,4,5-Trimethoxystyryl)-1H-Indole-3-Carboxamide (PILA9)

Here, 80 µL (0.46 mmol) of chlorosulfonyl isocyanide was added to a solution of 200 mg (0.62 mmol) of (*Z*)-1-Methyl-5-(3,4,5-trimethoxystyryl)-1*H*-indole in 10 mL of diethyl ether at 0 °C and under an Ar atmosphere. After 48 h at room temperature, the reaction is poured onto ice and extracted with ethyl acetate, washed with brine until neutral pH. The organic layers were dried over anhydrous Na_2_SO_4_, filtered, and evaporated in vacuo to yield 123 mg, which were crystallized in dichloromethane/ethyl ether to yield 80 mg (35%) of (*Z*)-1-Methyl-5-(3,4,5-trimethoxystyryl)-1*H*-indole-3-carboxamide (**PILA9**) as a light pink solid. M.p. (CH_2_Cl_2_/Et_2_O): 164–165 °C. IR (KBr): 3454, 3336, 1641, 1602, 1577, 1034. ^1^H-NMR (CDCl_3_): 3.63 (6H, s); 3.81 (3H, s); 3.83 (3H, s); 6.52 (1H, d, *J* = 12.4 Hz); 6.53 (2H, s); 6.72 (1H, d, *J* = 12.4 Hz); 7.21 (1H, d, *J* = 8.4 Hz); 7.25 (1H, d, *J* = 8.4 Hz); 7.62 (1H, s); 7.88 (1H, s). ^13^C-NMR (CDCl_3_): 33.4 (CH_3_); 55.8 (CH_3_ *2); 60.9 (CH_3_); 105.9 (CH *2); 109.3 (CH); 110.0 (CH); 120.9 (CH); 124.0 (CH); 125.4 (C); 128.9 (CH); 130.6 (CH); 130.9 (C); 132.4 (C); 133.4 (C); 136.5 (CH); 137.0 (C); 152.9 (CH *2); 166.8 (C). HRMS: 367.1652 calculated for C_21_H_23_N_2_O_4_+, found 367.1657 (M + H^+^).

### 4.4. Ensemble Docking Studies

Ensemble docking studies to take into account the protein flexibility were carried out with 81 models of complexes of tubulin with structurally diverse colchicine site ligands as previously described [9]. Briefly, dockings for each ligand were run in parallel with AutoDock 4.2 [58] applying a grid spacing of 0.375 Å and a Lamarckian genetic algorithm (LGA) for a maximum of 2.5 × 10^6^ energy evaluations 100–300 times, 150 individuals, and a maximum of 27,000 generations and with PLANTS [59] using the chemplp scoring function and a search speed of 1 with default settings and 10 runs. For each virtual ligand 810 poses were obtained with PLANTS and between 500 and 4000 poses with AutoDock. In-house KNIME pipelines [60] were applied to automatically locate all the retrieved poses within the colchicine subzones in tubulin [9]. Z-scores were calculated for the programs’ docking scores and those poses with the best consensus scores were selected as the docking results. Docking results were analyzed with Chimera [61], Marvin [62], OpenEye [63], and JADOPPT [64].

### 4.5. Cell Proliferation Assay

OCCLs were seeded into 96-well plates (4 × 10^3^ cells/well) and were treated or not with different concentrations of **PILA9** (1, 2.5, 5, 10, 25, and 50 nM) for 24, 48, or 72 h. Cell proliferation was determined using 3-(4,5-dimethylthiazol-2-yl)-2,5-diphenyl-2*H*-tetrazolium bromide (MTT) (Sigma-Aldrich, St. Louis, MO, USA). MTT salt was dissolved in PBS (5 mg/mL) and 10 µL of this salt per well was added to the cells. After 1 h of incubation, the medium was aspirated, and formazan crystals were dissolved in DMSO (100 µL/well). Absorbance was measured at 570 nm in a plate reader (Ultra Evolution, Tecan). The half maximal inhibitory concentration (IC_50_) was calculated using GraphPad Prism (version 9.0.1 for Mac).

### 4.6. Cell Cycle Analysis

OCCLs were treated with **PILA9** for 24, 48, and 72 h. After that, they were fixed in 70% ethanol and stored at 4 °C for later use. Cells were then rehydrated with PBS, stained with 50 µg/mL propidium iodide (PI) (Sigma-Aldrich, St. Louis, MO, USA), and treated overnight with 100 µg/mL RNase A in the dark (Sigma-Aldrich, St. Louis, MO, USA). Cell cycle profiles were analyzed by flow cytometry using BD Accuri™ C6 Plus Flow Cytometer (BD Biosciences, Haryana, Haryana, India). Data were analyzed with BD Accuri™ C6 Software (version 1.0.264.21).

### 4.7. Apoptosis Assay

OCCLs were treated with microtubule destabilizing drugs and/or LBH for 72 h and then stained with propidium iodide (PI) and annexin V-fluorescein isothiocyanate, using FITC Apoptosis Detection Kit CE (Immunostep, Salamanca, Spain). The percentage of apoptotic cells was determined by flow cytometry. The synergism of the combination was determined using Compusyn Software (version 1.0 for Windows, ComboSyn, Inc., Paramus, NJ, USA), which is based on the Chou-Talalay method [65] and calculates a combination index (CI) with the following interpretation: CI  >  1: antagonistic effect; CI  =  1: additive effect; CI  <  1 synergistic effect.

### 4.8. Western Blot

Cells were resuspended in lysis buffer (50 mM Tris-HCl pH 7.4, 130 mM NaCl, 1 mM EDTA, 1% Triton X-100) containing protease inhibitors (Complete, Roche Applied Science, Indianapolis) and protein concentration was measured using the Bradford assay (#5000006, BioRad, Hercules, CA, USA). Protein samples (30 μg/lane) were subjected to SDS-PAGE and then transferred to PVDF membranes (Immobilon-PSQ PVDF Membrane, Merck Millipore, Darmstadt, Germany). After blocking, membranes were incubated with primary antibodies against the following proteins: α-tubulin (1:5000, T6199, Sigma-Aldrich, St. Louis, MO, USA), acetylated α-tubulin (1:1000, T7451-25UL, Sigma-Aldrich, St. Louis, MO, USA), and β-actin (1:10,000, A5441, Sigma-Aldrich, St. Louis, MO, USA). β-actin was used as the loading control. Goat Anti-Mouse IgG (H+L) DryLight^TM^ 680 Conjugated (1:10,000, 35518, Invitrogen, Waltham, MA, USA) was used as the secondary antibody. Immunoblots were incubated for 1 h at room temperature and developed using Odyssey infrared imaging system (LI-COR Biosciences, Lincoln, NE, USA). Protein expression levels were calculated using ImageJ Software (ImageJ2, version 2.3.0/1.53q).

### 4.9. Immunofluorescence

Protein detection by immunofluorescence was carried out on coverslips pretreated with 2 µg/mL fibronectin, where cells were allowed to adhere to for 3 h. Cells were fixed using 4% paraformaldehyde (Thermo Fisher Scientific, Waltham, MA, USA) solution in PBS, and permeabilized with 0.1% Triton X-100 for 10 min. Coverslips were blocked for 30 min with PHEM buffer (60 mM PIPES, 25 mM HEPES, 10 mM EGTA, 2 mM MgCl_2_, 2% BSA, 0.05% NaN_3_, pH 6.9) and after that incubated with anti-acetylated α-tubulin (1:1000, T7451-25UL, Sigma-Aldrich, St. Louis, MO, USA) or anti-α-tubulin primary antibody diluted in PHEM buffer for 2 h at room temperature. Next, cells were rinsed and incubated with secondary antibody coupled to Alexa Fluor 488 goat anti-mouse IgG (1:1000, #A21202), phalloidin-Alexa Fluor 568 (1:1000, #A12380), and Hoechst (1:1000) from Invitrogen (Waltham, MA, USA) for 30 min a 37 °C. Finally, coverslips were rinsed and mounted on slides using ProLong™ Glass Antifade Mountant (Thermo Fisher Scientific, Waltham, MA, USA) and images were obtained using a Leica THUNDER microscope fitted with specific laser/filter combinations optimized for the fluorochromes used.

### 4.10. Determination of Tubulin Acetylation by Flow Cytometry

For flow cytometry analysis, cells were washed with PBS and dissociated from culture plates with TrypLE™ Express enzyme (Gibco, Waltham, MA, USA). Next, approximately 5 × 10^5^ cells were harvested in cytometry tubes, centrifuged for 3 min at 1800 rpm, and fixed with Buffer FIX (00-8222-49, Thermo Fisher Scientific, Waltham, MA, USA) for 10 min. Cells were then washed with PBSst (PBS with 0.5% FBS, 0.5% BSA, and 0.01% NaN_3_) and EDTA 2.5 mM before incubation with the primary antibody, which was performed for 30 min on ice. Anti-acetylated tubulin primary antibody was diluted in Buffer PERM 1x (00-833-56, Thermo Fisher Scientific, Waltham, MA, USA) at a 1:100 dilution. After one wash with PBSst, cells were incubated for 30 min in the dark with goat anti-mouse Alexa Fluor 488 secondary antibody (Invitrogen, Waltham, MA, USA) at a 1:1000 dilution. Then, cells were resuspended in PBSst+EDTA 2.5 mM, and acetylated tubulin relative levels were evaluated by flow cytometry (BD FACSAria™ III Sorter, BD Biosciences, Haryana, Haryana, India). Data were analyzed using FlowJo Software (version 10 for Windows).

### 4.11. Statistical Analysis

Differences between the results obtained from treated and non-treated cells were assessed for statistical significance using Student’s unpaired two-tailed *t*-test with Jamovi (version 2.2.5 for Mac). ANOVA with Tukey’s post-hoc test was used when more than two groups were compared. Data are presented as mean ± standard deviations. Statistical significance was concluded for values of *p* ≤ 0.05).

## 5. Conclusions

Altogether, our results strongly suggest the combination of a microtubule-destabilizing agent together with HDACi, such as Panobinostat, could represent a therapeutic strategy against ovarian cancer, especially for chemo-resistant tumors that do not respond to taxane therapy.

## Figures and Tables

**Figure 1 ijms-23-13019-f001:**
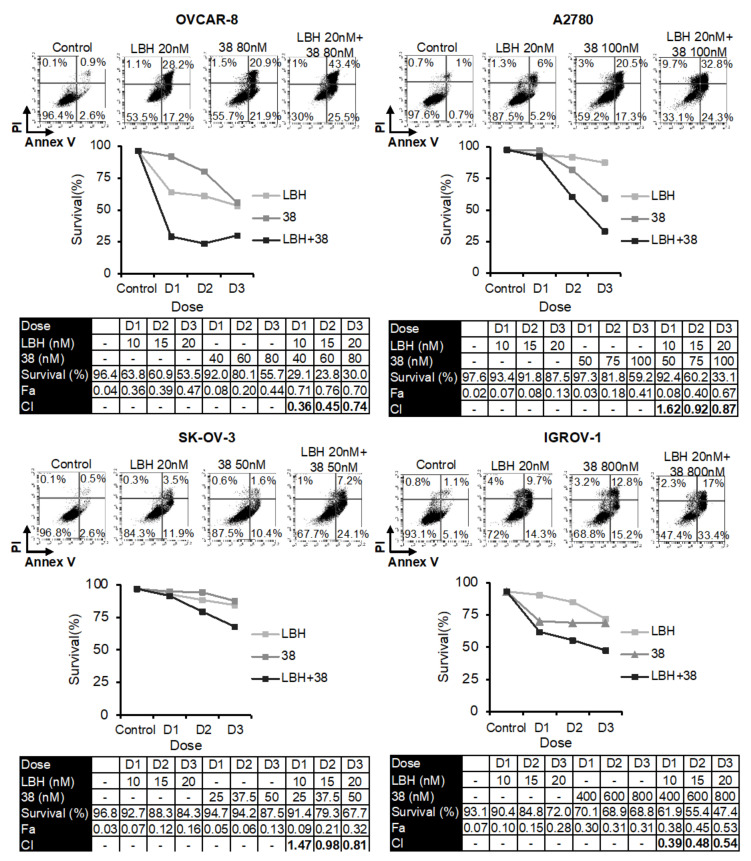
Synergistic effect of LBH and **38** in OCCLs. Cells were exposed for 72 h to the indicated concentrations of LBH and **38** at a constant ratio or vehicle control, and the percentage of apoptotic cells was assessed by flow cytometry (after cell staining with annexin V and propidium iodide). CI values, calculated using Compusyn Software, are shown.

**Figure 2 ijms-23-13019-f002:**
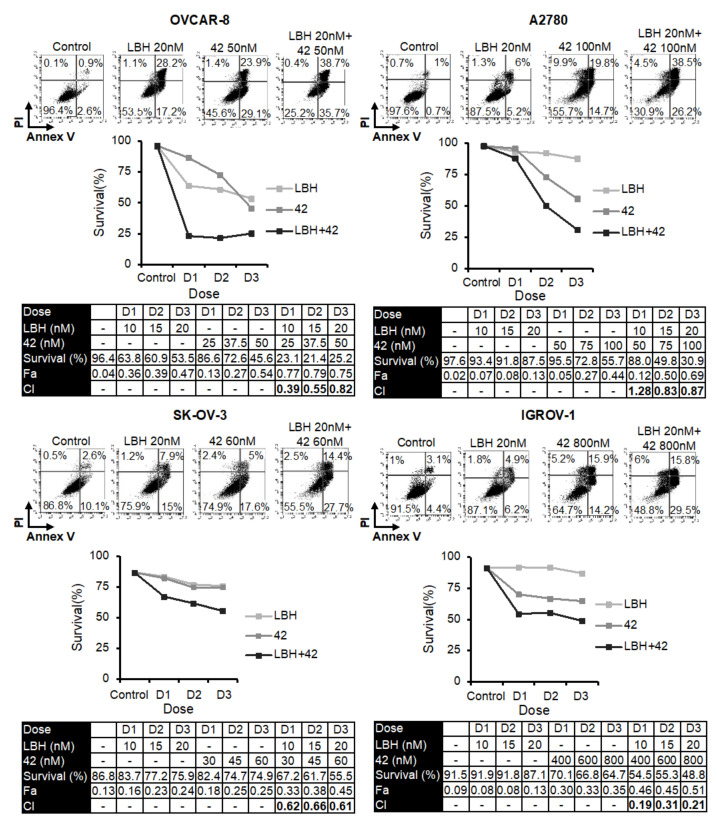
Synergistic effect of LBH and **42** in OCCLs. Cells were exposed for 72 h to the indicated concentrations of LBH and **42** at a constant ratio or vehicle control, and the percentage of apoptotic cells was assessed by flow cytometry (after cell staining with annexin V and propidium iodide). CI values, calculated using Compusyn Software, are shown.

**Figure 3 ijms-23-13019-f003:**
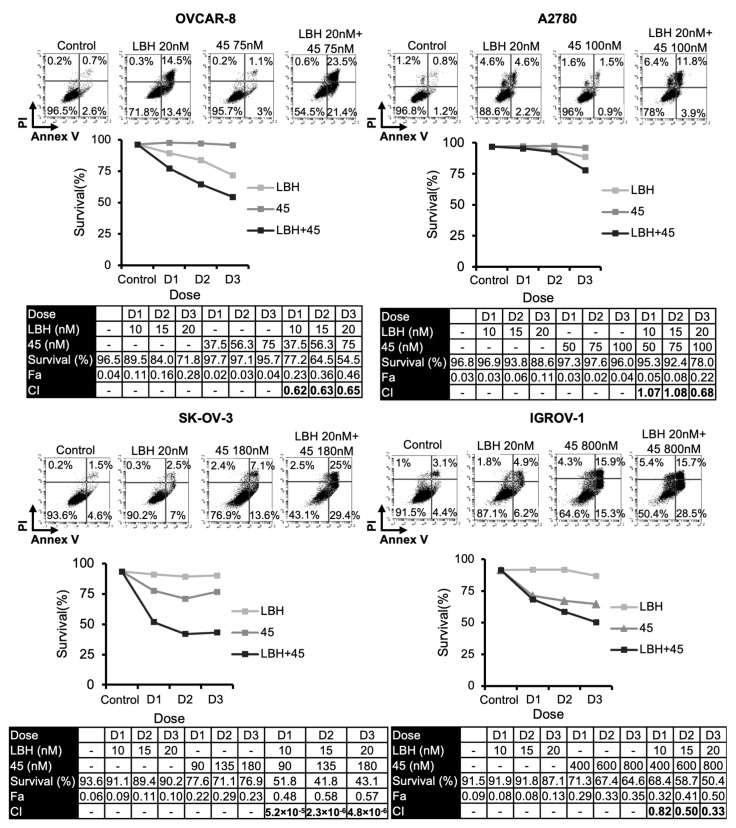
Synergistic effect of LBH and **45** in OCCLs. Cells were exposed for 72 h to the indicated concentrations of LBH and **45** at a constant ratio or vehicle control, and the percentage of apoptotic cells was assessed by flow cytometry (after cell staining with annexin V and propidium iodide). CI values, calculated using Compusyn Software, are shown.

**Figure 4 ijms-23-13019-f004:**
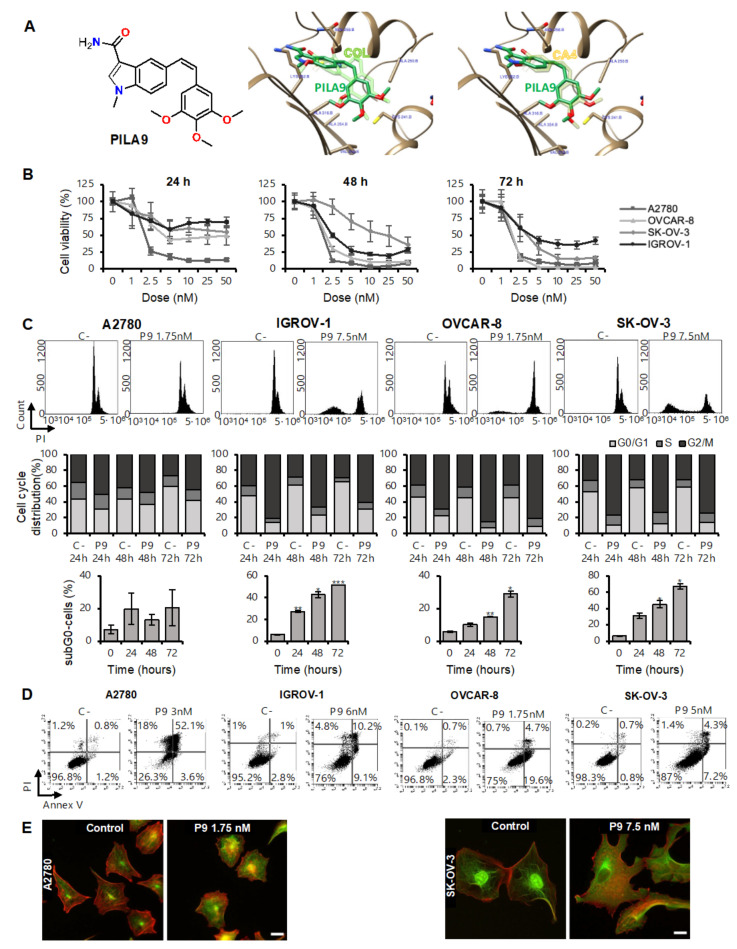
Effect of **PILA9** in proliferation, cell cycle distribution, apoptosis, and microtubule network in OCCLs. (**A**) Left panel: chemical structure of **PILA9**. Right panel: Consensus docking pose of the indolecombretastatin **PILA9** at the colchicine site of tubulin. Colchicine (COL)/Combretastatin A4 (CA4) is also shown for comparison. (**B**) Cell viability after treatment with the indicated doses of **PILA9** for 24, 48, and 72 h. (**C**) Top panel, cell cycle profile after **PILA9** treatment for 72 h. Middle panel, cell cycle distribution of OCCLs in the absence of treatment (C-) or after **PILA9** treatment for 24, 48, and 72 h, excluding the sub-G0 population. Bottom panel, percentage of death cells (subG0) after 24, 48, or 72 h of treatment with **PILA9**. (**D**) Dot plots showing alive cells (annexin V−/PI−), apoptotic (annexin V+), and necrotic (PI+) cells after 72 h of **PILA9** treatment. (**E**) Effect of **PILA9** on microtubule network. A2780 and SK-OV-3 were treated or not with **PILA9** for 24 h and α-tubulin (green) and actin (red) levels were observed by immunofluorescence. C-: negative control (untreated cells). Data are the mean of three independent experiments. Error bars represent the SD (*** *p* < 0.001; ** *p*  <  0.01, * *p*  <  0.05). Scale bar: 10 µm.

**Figure 5 ijms-23-13019-f005:**
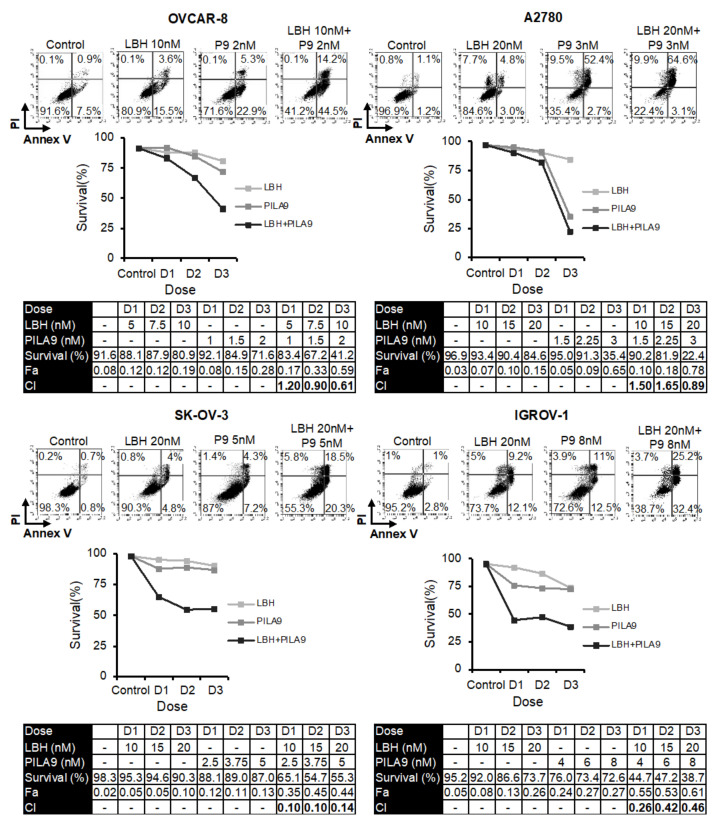
Synergistic effect of LBH and **PILA9** in OCCLs. Cells were exposed for 72 h to the indicated concentrations of LBH and **PILA9** at a constant ratio, and the percentage of apoptotic cells was assessed by flow cytometry (after cell staining with annexin V and propidium iodide). CI values, calculated using Compusyn Software, are shown.

**Figure 6 ijms-23-13019-f006:**
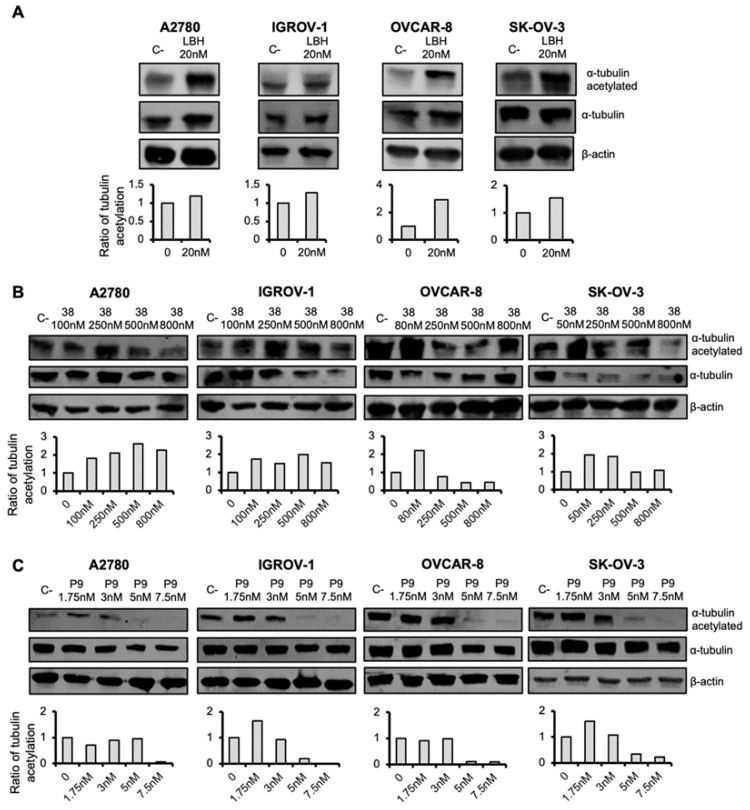
Effect of LBH and microtubule-destabilizing agents on tubulin acetylation. OC cells were treated with the indicated concentrations of LBH (**A**), **38** (**B**), or **PILA9** (**C**) for 24 h or left untreated (C-) and the levels of total α-tubulin and acetylated α-tubulin were detected by western blot. β-actin was used as a loading control. Protein levels were quantified using ImageJ. Graphs represented the normalized ratios of acetylated α-tubulin over total α-tubulin, using β-actin levels to normalize. Untreated cells (C-) levels were taken as 100.

**Figure 7 ijms-23-13019-f007:**
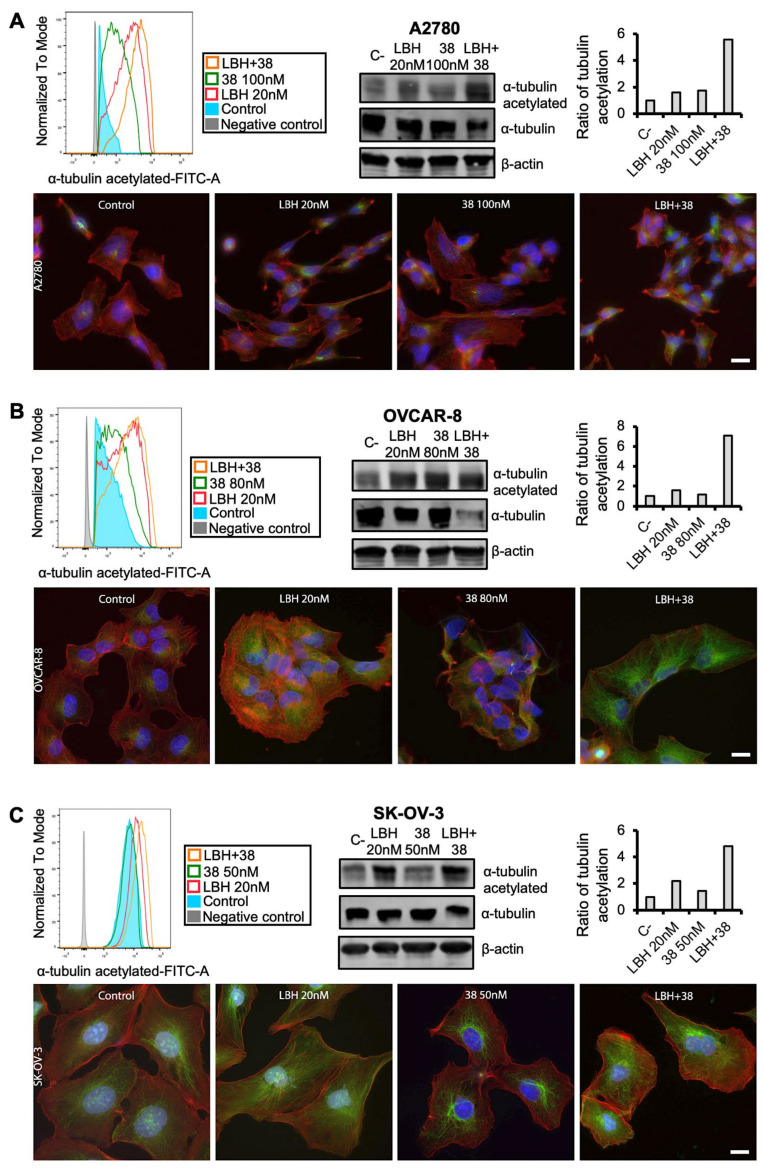
The combination of LBH and **38** induces increases α-tubulin acetylation. (**A**) A2780, (**B**) OVCAR-8 or (**C**) SK-OV-3 cell lines were treated with the indicated for 24 h and the acetylation of α-tubulin was detected by flow cytometry (**left panel**), western blot (**right panel**), and immunofluorescence (**bottom panel**). β-actin was used as a loading control. Protein levels were quantified using ImageJ. Graphs represented the normalized ratios of acetylated α-tubulin over total α-tubulin, using β-actin levels to normalize. Untreated cells (C-) levels were taken as 100. Acetylated tubulin is shown in green and actin in red. Nuclei were stained with Hoechst (blue). Scale bar: 10 µm.

**Figure 8 ijms-23-13019-f008:**
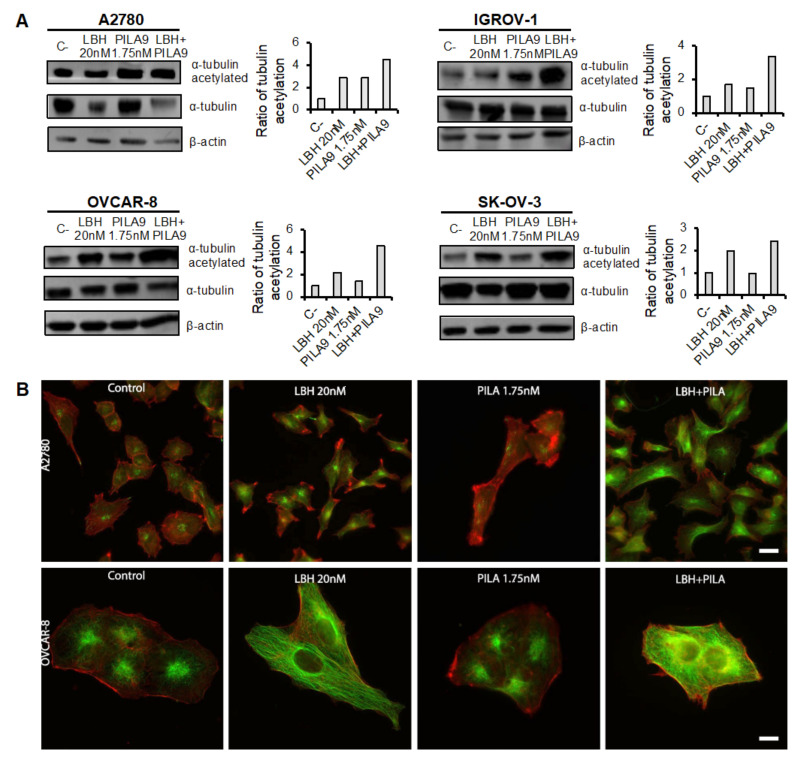
The combination of LBH and **PILA9** induces strong acetylation of α-tubulin. (**A**) OCCLs were treated or not with LBH (20 nM) and/or **PILA9** (1.75 nM) for 24 h and the acetylation of α-tubulin was observed by western blot. β-actin was used as a loading control. Protein levels were quantified using ImageJ Graphs representing the normalized ratios of acetylated α-tubulin over total α-tubulin, using β-actin levels to normalize. Untreated cells (C-) levels were taken as 100. (**B**) A2780 and OVCAR-8 cells were treated or not with LBH and/or **PILA9** and α-tubulin acetylated (green) and actin (red) were observed by immunofluorescence. Scale bar: 10 µm.

**Table 1 ijms-23-13019-t001:** IC50 values for MDA. Best-fit values for IC50 values and interval in which IC50 is included. These values were calculated using GraphPad Prism software (version 9).

OCCL	A2780	IGROV-1	OVCAR-8	SK-OV-3
**MDA**	IC50 (95% CI)	IC50 (95% CI)	IC50 (95% CI)	IC50 (95% CI)
**PILA9**	1.49 nM (1.12–1.97 nM)	6.43 nM (4.40–9.47 nM)	1.37 nM (0.92–1.96 nM)	3.34 nM (2.61–6.26 nM)
**38**	67.75 nM (40.95–111.5 nM)	248.6 nM (147.0–426.5 nM)	74.51 nM (57.57–96.05 nM)	46.31 nM (25.36–82.01 nM)
**42**	42.04 nM (29.21–59.33 nM)	400.1 nM (253.0–648.8 nM)	37.09 nM (28.05–48.66 nM)	7.60 nM (3.97–13.26 nM)
**45**	104.1 nM (69.70–153.6 nM)	492.1 nM (354.9–679.4 nM)	48.44 nM (34.55–66.31 nM)	47.91 nM (19.27–105.2 nM)

## Data Availability

Not applicable.

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
