# Peer review of "Panobinostat Synergistically Enhances the Cytotoxicity of Microtubule Destabilizing Drugs in Ovarian Cancer Cells"

_ijms, 2022, doi:10.3390/ijms232113019_

Round 1
Reviewer 1 Report
As resistant to chemotherapeutic drug is very common in Ovarian cancer (OC), there is an urgent need to explore and develope new and better therapeutic drugs. Toward this aim, these authors have previously developed three Microtubule Destabilizing Sulfonamides (MDS) for treatment of OV. In this study, they also identified a new and potent MDS, designated PILA9, that at very low doses suppresses growth of several OV cell lines in culture associated with induction of apoptosis. Recent studies by others have also identified HDACi that cooperate with MDS to increase apoptotic and growth suppression index of OV cell lines. Here, they showed that MDSs, including PILA9, cooperate with HDACi inhibitor Panobinostat (LBH) to further inhibit growth of OV cell lines. Moreover, LBH and MDS shown to increase the level of tubulin acetylation, leading to higher apoptosis and growth suppression in culture of OV cell lines. Overall, this study has identified a new MDS compound and shown to cooperate with LBH to exhibit highest inhibitory activity. The results are interesting, but may require additional experiments to justify the conclusions.
Major comments
Figure 1-3 and 5. It seems that all experiments in these figures only performed once and may need at least three times to draw statistical conclusions, especially survival and apoptosis analysis. It also lacks the vehicle control group.
As the new compound P9 claims to have higher tumor inhibitory activity, it would be appropriate to provide a table comparing its IC50 with other MDSs.
In figure 4H, it would be important to also include colchicine as positive control. It would also helpful to explain why cells become smaller after treatment with H9.
In figure 6c, higher doses of P9 compound causing downregulation of α-tubulin acetylation. However, the significance is not clear and needs to be clarified.
Figure7B/c, the total tubulin level is not equal and the blot may need to be replaced with better image.
Minor points
In line 203, Figure 6 should be Figure 6A.
Last figure should be figure 8 and not figure 5.
Author Response
As resistant to chemotherapeutic drug is very common in Ovarian cancer (OC), there is an urgent need to explore and develope new and better therapeutic drugs. Toward this aim, these authors have previously developed three Microtubule Destabilizing Sulfonamides (MDS) for treatment of OV. In this study, they also identified a new and potent MDS, designated PILA9, that at very low doses suppresses growth of several OV cell lines in culture associated with induction of apoptosis. Recent studies by others have also identified HDACi that cooperate with MDS to increase apoptotic and growth suppression index of OV cell lines. Here, they showed that MDSs, including PILA9, cooperate with HDACi inhibitor Panobinostat (LBH) to further inhibit growth of OV cell lines. Moreover, LBH and MDS shown to increase the level of tubulin acetylation, leading to higher apoptosis and growth suppression in culture of OV cell lines. Overall, this study has identified a new MDS compound and shown to cooperate with LBH to exhibit highest inhibitory activity. The results are interesting, but may require additional experiments to justify the conclusions.
Major comments
Figure 1-3 and 5. It seems that all experiments in these figures only performed once and may need at least three times to draw statistical conclusions, especially survival and apoptosis analysis. It also lacks the vehicle control group.
In this work, CompuSyn software was employed (http://www.combosyn.com). CompuSyn is based on the Chou-Talalay theory (DOI: 10.1158/0008-5472.CAN-09-1947). In this study, Chou indicates that synergism follows a physical-chemical mass-action law and as such is not a statistical issue. For this reason, it is recommended to determined synergism by calculation of the combination indices’, based on different drug doses at constant ratios, not with p-values. Indeed, a combined effect greater than individual effect does not necessarily indicate synergism and can present a significative p-value. In this case, it can be a result of an additive effect.
The significance of the data obtained using the Chou-Talalay model can be evaluated by an r value, that indicates the fitting of the data to the mass-action law (DOI: 10.1016/j.crphar.2022.100110; DOI: 10.1124/pr.58.3.10). In our case, r values were between 0.85 and 0.99 in all the combinations assayed, which indicates a confident fitting of the data to the mass-action law.
Nevertheless, all the experiments were repeated more than once at several doses and CIs obtained always indicated synergisms. Regarding vehicle control, we always added the solvent in which our reagents were dissolved, vehicle control group corresponded to “0” or “control” in the Figures. We have clarified this point in the figures and figure legends.
As the new compound P9 claims to have higher tumor inhibitory activity, it would be appropriate to provide a table comparing its IC50 with other MDSs.
We thank the reviewer for this suggestion. A table containing IC50 values for the different MDA has been added to the manuscript (Table 1).
In figure 4H, it would be important to also include colchicine as positive control. It would also helpful to explain why cells become smaller after treatment with H9.
A new supplementary figure that includes the chemical structure of colchicine, combretastatin A4 and PILA9 has been included.
Control cell sizes previously shown in Figure 4 were not representative of the population. We have replaced this picture by other more appropriate. Thanks.
In figure 6c, higher doses of P9 compound causing downregulation of α-tubulin acetylation. However, the significance is not clear and needs to be clarified.
At high doses of P9, MTs get completely disassembled. It is well known that only polymerized (and not free) tubulin undergoes K40 acetylation. In fact, K40 acetylation only happens in the lumen of microtubules (DOI: 10.1016/j.cub.2017.10.044), contributing to the stabilization of the polymer by reducing the degree of freedom of the αK40 loop (DOI: 10.1073/pnas.1900441116). Our experiments suggest that, at low doses (1.75nM), the predominant effect of P9 could be to hinder HDAC6 interaction with MTs, maybe by “loosening” the microtubule structure. This would increase MT acetylation, as it can be observed in our data (Fig. 8A). Conversely, at high doses, the organization of the MT tree is severely affected. This agrees with our observations that, at low doses, P9 has a very modest effect on MT organization (see Fig. 8B, OVCAR-8 cells at 1.75nM P9), whereas MTs are mainly disorganized at high doses (see Fig. 4E, SK-OV-3 panel with 7.5nM P9).
This information has been incorporated into the discussion
Figure7B/c, the total tubulin level is not equal and the blot may need to be replaced with better image.
In this work, β-actin was used as the most appropriate loading control to avoid spurious effects due to the use of tubulin-interacting drugs. The reviewer is right in that total tubulin seems to decrease when some cell lines cells were treated with the combination 38-LBH, and this is an interesting observation. However, the tested combinations clearly increased É‘-tubulin acetylation (which is the aim of this figure) since graphs represented the normalized ratios of acetylated É‘-tubulin over total É‘-tubulin, using β-actin levels to normalize.
Minor points
In line 203, Figure 6 should be Figure 6A.
Changed, thank you.
Last figure should be figure 8 and not figure 5.
Changed, thank you.

Reviewer 2 Report
The manuscript by Ovejero-Sanchez et al. describes the use of an HDACi, Panobinostat, in combination with various tubulin inhibitors against ovarian cancer cells. The authors report cytotoxicity with the individual MDS and PILA9 tubulin inhibitors, but a synergistic effect leading to enhanced cytotoxicity when combined with Panobinostat. Further, PILA9 was significantly more cytotoxic than their previously published MDS tubulin inhibitors. This work may be especially useful for chemo-resistant tumors.
This work is suitable for publication in the International Journal of Molecular Sciences after consideration of the following points:
1. Line 92: ‘MTs’ is not previously defined.
2. Line 107: The compounds 38, 42, and 45 should be bolded for clarity, and consistency with formatting of reporting compounds.
3. Section 2.2 under Results would benefit from a figure comparing the structures of PILA9 to combretastatin A4, and possibly colchicine and other colchicine binding site inhibitors.
4. This reviewer would like to see more discussion of the docking results. What interactions are made with tubulin across the alpha/beta chains? How does this compare to combretastatin A4, colchicine, and other colchicine binding site inhibitors? Such analysis may help inform mechanistic information.
5. The docking studies in Materials and Methods also need more detail. What is the grid space? What receptor was used in AutoDock?
Overall, this paper is interesting and well written. It logically flows from previous work. It does suffer from some redundancy across the different sections, but that can be fixed easily. This reviewer recommends it for publication after minor revisions.
Author Response
The manuscript by Ovejero-Sanchez et al. describes the use of an HDACi, Panobinostat, in combination with various tubulin inhibitors against ovarian cancer cells. The authors report cytotoxicity with the individual MDS and PILA9 tubulin inhibitors, but a synergistic effect leading to enhanced cytotoxicity when combined with Panobinostat. Further, PILA9 was significantly more cytotoxic than their previously published MDS tubulin inhibitors. This work may be especially useful for chemo-resistant tumors.
This work is suitable for publication in the International Journal of Molecular Sciences after consideration of the following points:
- Line 92: ‘MTs’ is not previously defined.
We have defined the meaning in line 44. Thanks
- Line 107: The compounds 38, 42, and 45 should be bolded for clarity, and consistency with formatting of reporting compounds.
Compounds have been marked in bold type. Thanks
- Section 2.2 under Results would benefit from a figure comparing the structures of PILA9 to combretastatin A4, and possibly colchicine and other colchicine binding site inhibitors.
We have added a supplemental figure (Figure S1) with the structure of PILA9, combretastatin A-4, and colchicine.
Figure S1. Chemical structures of combretastatin A-4, PILA-9, and colchicine.
- This reviewer would like to see more discussion of the docking results. What interactions are made with tubulin across the alpha/beta chains? How does this compare to combretastatin A4, colchicine, and other colchicine binding site inhibitors? Such analysis may help inform mechanistic information.
The binding mode of PILA9 at the colchicine site of tubulin was studied by flexible docking studies and taking into account the protein conformational variability by using 82 protein structures with different binding sites, as previously described [9]. The docking programs are two frequently used which use different scoring functions. As a result of the docking process, several thousand poses were generated for each ligand which were automatically classified according to their occupation of the subsites of the colchicine domain. The scores of each individual pose were converted to Z scores in order to make comparisons between the two docking programs possible. The poses with the best scores for the two software were selected as the consensus pose.
PILA9 binds to zones A and B of the colchicine site in a similar disposition to combretastatin A-4, with a very similar arrangement of the two phenyl rings: the trimethoxyphenyl ring of both compounds sits in the A zone and the other aromatic system in zone 2, equally to the MDS. A close overlap of the trimethoxyphenyl ring of PILA9 with that of the X-ray structure of combretastatin A4 in complex with tubulin is observed. The trimethoxyphenyl ring inserts edgewise towards the surface of sheets S8 and S9 between the sidechains of Ala316β, Val318β, Ala354β, and covered by helices H7 and H8 and by the H7-H8 loop, contacting the sidechains of Cys241β, Leu242β, Leu248β, Ala250β, and Leu255β. The olefinic bridge is also placed similarly to that of combretastatin A-4 and the sulfonamide bridges of the MDS, packed against helix H8 between Leu255β and Leu248β in a hydrophobic pocket at the interdimer interface. The indole system overlaps as well with the phenyl ring of combretastatin A4 and the MDS, with the N-methyl replacing the methoxy groups of combretastatin A4 or the MDS. The indole ring lays behind helix H8, making carbonyl pi interactions with Asn258β and above the sidechains of Ala316β and the methylene groups of the sidechain of Lys352β. The carbonyl group of the carbamoyl group hydrogen bonds to the backbone NH of Val181α, in a similar way as the hydroxyl group of combretastatin A4 or the ketone of the tropolone of colchicine, while the amino group hydrogen bonds to the backbone carbonyl of Asn349β.
We have added this description to the Results section.
We have added in the Discussion a reference to the comparison of the binding modes of the MDS, PILA9 and those observed in the X-ray structures of combretastatin A4 and other colchicine site ligands:
“Molecular docking studies predict PILA9 binds to the colchicine site of tubulin, which likely underlies the microtubule alterations observed after treatment with this compound. The binding mode of PILA9 to the colchicine site of tubulin is similar to those of the MDS and also to the experimentally determined binding modes of other colchicine site ligands such as the combretastatin A4 or colchicine [9], thus suggesting a common mechanism of action mediated by tubulin binding.”
- The docking studies in Materials and Methods also need more detail. What is the grid space? What receptor was used in AutoDock?
We have used a grid spacing of 0.375 Å. As previously described, [9] we have used an ensemble of 81 X-ray structures of tubulin in complex with structurally different colchicine–site binding ligands to account for receptor flexibility. The pdb codes can be found in the supplemental material of reference [9]. We have modified ref [9] in the manuscript and described in more detail the docking procedure in the Materials and Methods section.
Overall, this paper is interesting and well written. It logically flows from previous work. It does suffer from some redundancy across the different sections, but that can be fixed easily. This reviewer recommends it for publication after minor revisions.

Round 2
Reviewer 1 Report
The authors have addressed my comments and this paper is now acceptable for publication.